# FLOWCYCLE: PURSUING CYCLE-CONSISTENT FLOWS FOR TEXT-BASED EDITING

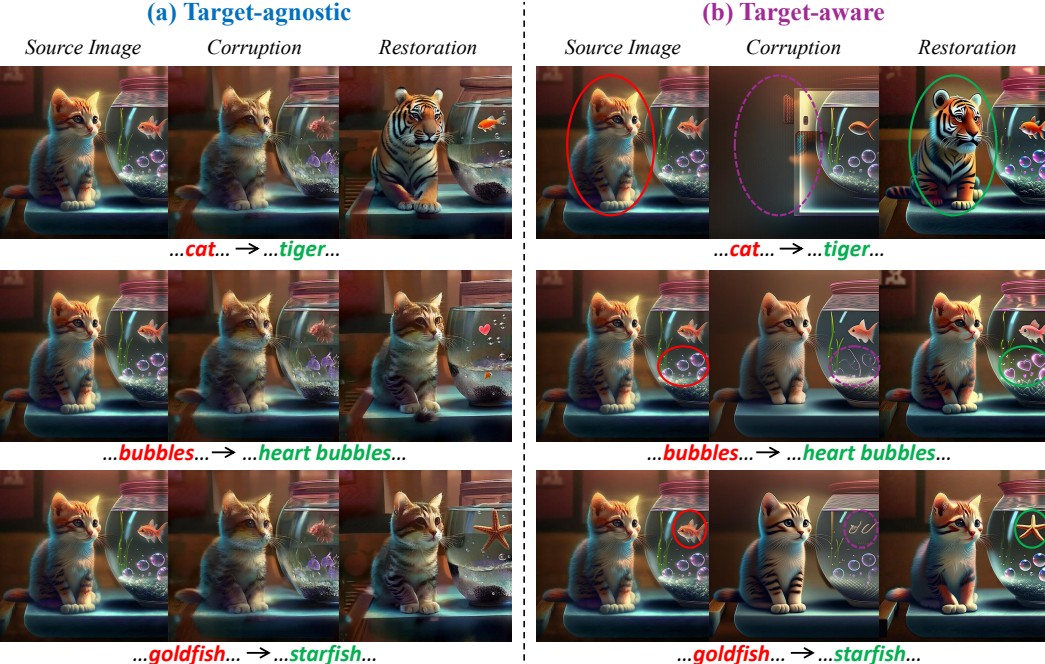

Figure 1: **Comparisons between target-agnostic and target-aware manner**. (a) Target-agnostic corruption equally destroys global content, which makes it difficult to restore the destroyed editing-irrelevant content well. (b) Target-aware corruption emphatically destroys editing-relevant content, which can better maintain editing-irrelevant content. Details about the visualization implementations and more cases are in the Appendix A.7.

## ABSTRACT

Recent advances in pre-trained text-to-image flow models have enabled remarkable progress in text-based image editing. Mainstream approaches always adopt a *corruption-then-restoration* paradigm, where the source image is first corrupted into an "intermediate state" and then restored to the target image under the prompt guidance. However, current methods construct this intermediate state in a *target-agnostic* manner, *i.e.*, they primarily focus on realizing source image reconstruction while neglecting the semantic gaps towards the specific editing target. This design inherently results in limited editability or inconsistency when the desired modifications substantially deviate from the source. In this paper, we argue that the intermediate state should be *target-aware*, *i.e.*, selectively corrupting editing-relevant contents while preserving editing-irrelevant ones. To this end, we propose **FlowCycle**, a novel inversion-free and flow-based editing framework that parameterizes corruption with learnable noises and optimizes them through a cycle-consistent process. By iteratively editing the source to the target and recovering back to the source with dual consistency constraints, FlowCycle learns to produce a target-aware intermediate state, enabling faithful modifications while preserving source consistency. Extensive ablations have demonstrated that FlowCycle achieves superior editing quality and consistency over state-of-the-art methods.

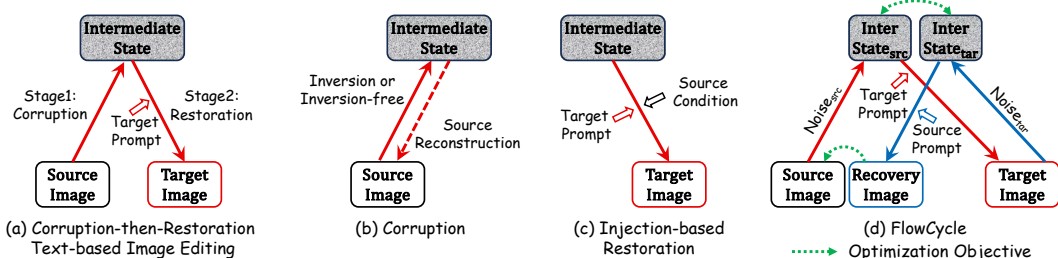

Figure 2: Corruption-then-restoration paradigm for existing flow-based image editing methods[1].

# 1 INTRODUCTION

Pre-trained text-to-image flow models (Esser et al., 2024; Labs, 2024) have recently proven strong generative capacity, which has gathered growing interest in adapting them for text-based editing (Xu et al., 2024; Rout et al., 2024; Kulikov et al., 2024). Unlike standard text-to-image generation, editing requires not only producing new content but also preserving certain source details. Specifically, given a source image and its description, the key challenge for text-based editing is to modify the image according to a target prompt while maintaining non-target content as the source, *i.e.*, *source consistency*. To achieve this goal, most existing flow-based editing approaches (Xu et al., 2024; Rout et al., 2024; Kulikov et al., 2024; Kim et al., 2025a) closely follow well-established diffusion-based methods, since both flow-based and diffusion-based (Ho et al., 2020; Song et al., 2020a) models share a similar generative principle, *i.e.*, they corrupt the clean image with progressive noise and then learning to reverse this process to generate new samples. Consequently, existing flow-based editing methods generally adopt a common ***corruption-then-restoration*** paradigm. As shown in Figure 2(a), this paradigm consists of two main stages:

1. **Corruption.** The source image is first transformed into an *intermediate state* (*i.e.*, the corrupted image), which allows flexible editing while maintaining the source consistency required to restore the source image. As shown in Figure 2(b), the representative approaches include *inversion-based* methods (Deng et al., 2024; Zhou et al., 2025) that perform ODE inversion (Song et al., 2020a) and subsequent improvements (Rout et al., 2024; Wang et al., 2024; Kim et al., 2025b) to enhance the source reconstruction accuracy. As well as *inversion-free* methods (Kulikov et al., 2024; Kim et al., 2025a; Mao et al., 2025) that directly sample noise and add it into the source image to improve efficiency.

2. **Restoration.** Starting from the intermediate state, the pre-trained flow model restores the image under the guidance of the target prompt. To further enhance the preservation of source consistency, many methods inject source conditions (*c.f.*, Figure 2(c)) during the restoration stage (Wang et al., 2024; Kim et al., 2025b; Rout et al., 2024; Zhou et al., 2025).

In that case, an *ideal intermediate state* of this paradigm should strike a balance between editability and preservation of source consistency. As shown in Figure 1(b), given a specific source-target prompt pair that aims to edit the cat (*e.g.*, ...cat... → ...tiger...), the corruption should be applied to the editing-relevant contents (*e.g.*, cat) to ensure faithful modifications, while the source information of editing-irrelevant contents (*e.g.*, fish tank) should be preserved to maintain reliable source consistency. However, existing methods all generally acquire the intermediate state in a ***target-agnostic*** manner, *i.e.*, they emphasize the reconstruction of the entire source image without considering the specific editing targets, relying solely on the prior knowledge (*e.g.*, source prompt and pre-trained model) to achieve this balance. As shown in Figure 1(a), such intermediate states reveal an approximate reconstruction of the source image, preserving unbiased global layout structures and semantic details. When restored with different target prompts focusing on distinct contents, this source-aware but target-agnostic design often leads to unsatisfactory results, especially when the editing target significantly deviates from the source image or the prior knowledge. For instance, as shown in the Figure 1(a), when the target is to change the shape of the bubble (*e.g.*, ...bubbles... → ...heart bubbles...), the target-agnostic intermediate state not only fails to reliably preserve editing-irrelevant contents (*e.g.*, cat and goldfish) but also conflict with editing-relevant

---

[1]We provide some justifications for summarizing existing methods into this paradigm in Appendix A.6.

contents (*e.g.*, `bubbles`), making them difficult to modify as desired. Meanwhile, even injecting additional source conditions during restoration cannot resolve the problem, since the intermediate state itself remains misaligned with the editing target.

Therefore, in this paper, we argue that the *ideal intermediate state* should be constructed in a ***target-aware*** manner, selectively corrupting editing-relevant contents while preserving editing-irrelevant ones based on specific editing targets to enable faithful and effective editing. To this end, we introduce **FlowCycle**, a novel inversion-free and flow-based framework that faithfully incorporates target awareness, enabling precise and reliable text-based image editing. Notably, such a target-aware state exhibits a natural symmetry between the source and target image: the editing-relevant regions should be interchangeable, while the editing-irrelevant regions should remain invariant across both. Motivated by this intuition, we construct a *cycle* between the source and target images, allowing the model to exploit this inherent symmetry and learn a target-aware corruption.

Specifically, instead of relying on fixed inversion or random noise sampling, our method parameterizes the corruption process with learnable noises through a cycled framework. As shown in Figure 2(d), it consists of three main steps. *1) Source to target* (*i.e.*, the red path of Figure 2(d)): the source image is first corrupted with a learnable source noise and then restored under the guidance of the target prompt, producing an initial target image that reflects the desired edit. *2) Target to source* (*i.e.*, the blue path of Figure 2(d)): the initial target image is also corrupted with another learnable target noise and then restored with the source prompt, yielding a recovery image that traces back to the source. *3) Cycle-consistent Optimization*: we impose two constraints to optimize the learnable noises — (i) align the recovery image to the source image, and (ii) align the corrupted source image with the corrupted target image. This joint regularization stabilizes the optimization and enables target-aware corruption. By iterating this process, the learned noises gradually produce a target-aware intermediate state (*c.f.*, Figure 1(b)), where the corrupted source image and the corrupted target image share an ideal role, *i.e.*, corrupt the editing-relevant content while maintaining the editing-irrelevant parts based on a specific editing target.

In summary, we made three contributions in this paper: 1) We identify a fundamental limitation in prevalent corruption-then-restoration editing methods: the intermediate state is acquired in a *target-agnostic* manner, which often leads to unsatisfactory results. 2) We propose the first *target-aware* corruption strategy and introduce FlowCycle, an inversion-free, flow-based framework that adaptively focuses corruption on editing-relevant content while preserving irrelevant content. 3) Extensive ablations have empirically shown the effectiveness of FlowCycle over state-of-the-art methods.

## 2 RELATED WORK

Existing text-based image editing methods largely converge on the *corruption-then-restoration* paradigm, improving editing performance at either the corruption or the restoration stage. While a few alternative approaches, such as unified diffusion models (Fu et al., 2025; Xiao et al., 2025) or MLLM-driven models (Deng et al., 2025; AI et al., 2025), also contributed to the field, this paper focuses primarily on methods within this framework due to its broad adoption and foundational role.

**Inversion-based Corruption.** Methods in this category construct the intermediate state from the source image via ODE inversion (Song et al., 2020a; Deng et al., 2024; Zhou et al., 2025), which allows the preservation of structural and semantic details for reconstruction (*i.e.*, restore with an empty prompt). However, the iterative inversion process is prone to error accumulation (Huberman-Spiegelglas et al., 2024; Mokady et al., 2023), which not only degrades the fidelity of source information but also weakens the reliability of subsequent editing. Thus, recent works focus on improving inversion accuracy (Rout et al., 2024; Wang et al., 2024; Kim et al., 2025b), enabling more faithful reconstruction of the source image and providing a stronger foundation for high-quality editing.

**Inversion-free Corruption.** To alleviate the high computational overhead caused by ODE inversion, inversion-free methods (Kulikov et al., 2024; Kim et al., 2025a; Mao et al., 2025) bypass the inversion step by directly sampling random noise to the source image to construct the intermediate state. Compared with inversion-based strategies, these methods achieve significantly faster corruption. However, they remain highly sensitive to noise level (Meng et al., 2021) and hyperparameter choices (Kulikov et al., 2024; Kim et al., 2025a). Excessive corruption risks erasing key source

information, while insufficient perturbation limits editability. Thus, the central challenge lies in striking the right balance between editability and faithful preservation of source consistency.

**Injection-based Restoration.** Solely relying on the intermediate state for restoration often results in partial information loss and inconsistency in the edited image (Huberman-Spiegelglas et al., 2024). To address this, a number of methods propose to explicitly inject source information throughout the restoration process, ensuring stronger alignment between the edited image and the source image. This can be achieved through additional model inputs, *e.g.*, optimized prompt (Mokady et al., 2023) and latent (Wu & De la Torre, 2023; Huberman-Spiegelglas et al., 2024). Intermediate model features (Wang et al., 2024; Kim et al., 2025b) derived from the source image, *e.g.*, attention maps (Hertz et al., 2022; Tumanyan et al., 2023). And enhanced model outputs (Rout et al., 2024; Zhou et al., 2025) such as predicted noise or velocity. Nevertheless, without a properly constructed and target-aware intermediate state, such injection strategies still offer limited improvements.

**Cycle Consistency in Generation.** Based on the domain transfer ability of diffusion models, the well-recognized cycle consistency (Zhu et al., 2017) has also been explored, particularly in image-to-image translation tasks, where consistency regularization is enforced through translation cycles between source and target domains. Such methods typically require images from two specific domains for training or optimization(Sasaki et al., 2021; Xu et al., 2023; Lee et al., 2023). While recent work has attempted to adapt cycle consistency to image editing (Beletskii et al., 2025), their "cycle" still focuses solely on source inversion and reconstruction. In contrast, our work leverages cycle consistency to enable a novel *target-aware* corruption during text-based editing, where the cycle is established between source and target to explicitly align editing-relevant and irrelevant contents.

## 3 METHOD

### 3.1 PRELIMINARIES: FLOW MATCHING (FM)

Flow Matching (FM) models (Lipman et al., 2022; Liu et al., 2022) are trained to fit a velocity field $u_t(x_t)$ on time $t \in [0, 1]$ such that they can transport data from one distribution (*e.g.*, normal distribution $\pi_1$) to another distribution (*e.g.*, images distribution $\pi_0$). Specifically, given a data point $x_1 \sim \pi_1$, the transport path can be simulated by solving an ordinary differential equation (ODE):

$$dx_t = u_t(x_t)dt, \tag{1}$$

which converts $x_1$ into $x_0 \sim \pi_0$. To match the underlying velocity field $u_t(x_t)$, we can train a velocity prediction network (Chen et al., 2018) $v_\theta(x_t, t)$ by minimizing the following FM loss:

$$\mathcal{L}_{FM} = \mathbb{E}_{x_t,t}\|v_\theta(x_t, t) - u_t(x_t)\|^2. \tag{2}$$

However, the ground-truth velocity field $u_t(x_t)$ is not accessible. Alternatively, prior studies (Lipman et al., 2022; Liu et al., 2022) propose to optimize the approximated conditional FM loss:

$$\mathcal{L}_{CFM} = \mathbb{E}_{x_0,p_t(x_t|x_0),t}\|v_\theta(x_t, t) - u_t(x_t|x_0)\|^2, \tag{3}$$

where $p_t(x_t|x_0)$ is usually defined as the optimal transport (McCann, 1997) conditional probability path, *i.e.*, $p_t(x_t|x_0) = \mathcal{N}(x_t|(1-t)x_0, t^2 I)$ while $I$ is the identity matrix. Since $\pi_1$ is the standard normal distribution, we can sample $x_1 \sim \pi_1$ and get:

$$x_t = (1-t)x_0 + tx_1 \Rightarrow u_t(x_t|x_0) = \frac{dx_t}{dt} = x_1 - x_0. \tag{4}$$

By substituting $u_t(x_t|x_0) = x_1 - x_0$ into Eq. (3), the final training objective becomes:

$$\mathcal{L}_{CFM} = \mathbb{E}_{x_0 \sim \pi_0, x_1 \sim \pi_1, t}\|v_\theta(x_t, t) - (x_1 - x_0)\|^2. \tag{5}$$

In the inference stage, we can start from a randomly sampled Gaussian noise $x_1 \sim \pi_1$ and then integrate to generate a clean image: $x_0 = x_1 - \int_1^0 v_\theta(x_t, t)dt$.

### 3.2 GENERAL CORRUPTION-THEN-RESTORATION EDITING PARADIGM

Given a source image $x_0^{src}$, source prompt $c_{src}$, and target prompt $c_{tar}$, the paradigm has two stages:

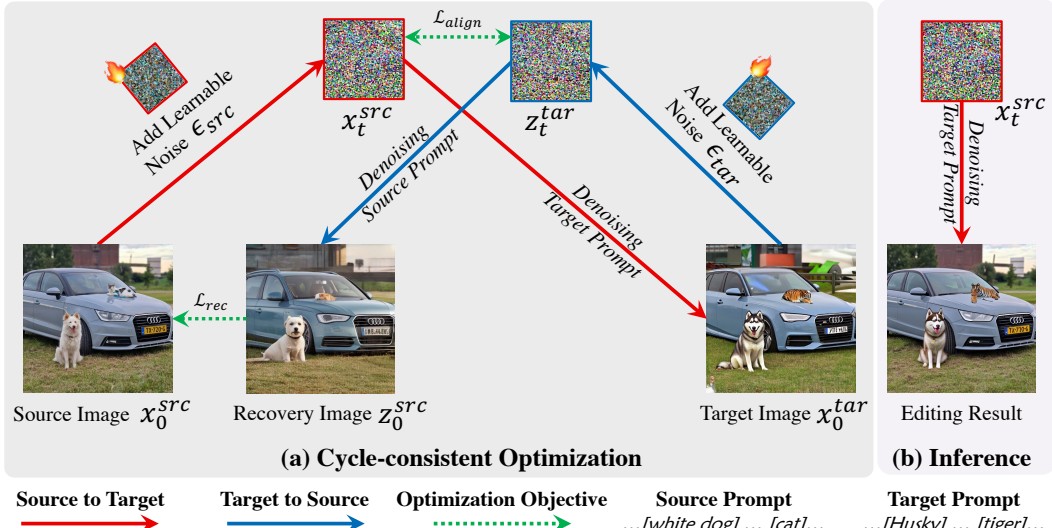

Figure 3: **Framework of FlowCycle**. We rectify the corruption process by optimizing two learnable noises (*i.e.*, $\epsilon_{src}$ and $\epsilon_{tar}$) under the cycle-consistent constraints of whole editing. After a few rounds of optimization, the quality of the editing result $x_0^{tar}$ can be significantly improved.

1) **Corruption**: Convert the source image $x_0^{src}$ into an editable noisy intermediate state $x_t^{src}$:

$$x_t^{src} = \psi(x_0^{src}, t, c_{src}), \quad t \in (0, 1], \tag{6}$$

where $\psi$ is a corruption function such as directly adding random noise (inversion-free) or calculating an inversion (inversion-based). The principle of both corruptions is obtaining an intermediate state $x_t^{src}$ while maintaining the information of the source image $x_0^{src}$. In that case, starting from $x_t^{src}$ can benefit *source consistency*, *i.e.*, the editing result can maintain the editing-irrelevant part. It is worth noting that current corruption is target-agnostic, *i.e.*, the noisy image $x_t^{src}$ is fixed for different editing targets. In contrast, our motivation is to find an ideal target-aware $x_t^{src}$ that can destroy the editing-relevant part while maintaining the editing-irrelevant part.

2) **Restoration**: After getting $x_t^{src}$, the most straightforward approach is to take it as the starting point of restoration directly (*i.e.*, $x_t^{tar} = x_t^{src}$) and denoising it under the guidance of the target prompt $c_{tar}$. However, only relying on the starting point that contains the source image's information is not enough for *source consistency*. Thus, existing methods inject a drift item $\Delta_t$ (containing the source image's information) during the denoising:

$$dx_t^{tar} = \xi(v_\theta, x_t^{tar}, t, c_{tar}, \Delta_t)dt, \tag{7}$$

where $\xi$ represents different injection functions. For instance, adding $\Delta_t$ to the middle attention maps, predicted velocity $v_\theta$, or input $x_t^{tar}$. The principle is enhancing the *source consistency* of the final editing result by injecting a source-relevant drift item $\Delta_t$ during denoising. This drift item can rectify the denoising path and help the final editing result maintain more editing-irrelevant parts.

### 3.3 FLOWCYCLE

**Overview**. As shown in Figure 3 (a), FlowCycle contains three steps: 1) Source to Target: We first add a parameterized learnable noise $\epsilon_{src}$ (initialized as a Gaussian noise) to the source image $x_0^{src}$ and get the intermediate state $x_t^{src}$. After getting $x_t^{src}$, we leverage the pretrained flow matching model to denoise it conditioned on the target prompt $c_{tar}$. The denoised result is the target image $x_0^{tar}$. 2) Target to Source: After getting the initialized target image $x_0^{tar}$, we add another parameterized learnable noise $\epsilon_{tar}$ (initialized as a Gaussian noise) to it and then get the noisy image $z_t^{tar}$. After that, we use the flow matching model conditioned on the source prompt $c_{src}$ to denoise it into a recovery image $z_0^{src}$. 3) Cycle-consistent Optimization: After (1) and (2), we calculate Mean Squared Error (MSE) loss between $x_t^{src}$ and $z_t^{tar}$ as well as the MSE loss between $x_0^{src}$ and $z_0^{src}$. Then we optimize the parameterized $\epsilon_{src}$ and $\epsilon_{tar}$ to decrease the loss. By repeating the three steps, $\epsilon_{src}$ and $\epsilon_{tar}$ are iteratively optimized. In the inference stage (*c.f.*, Figure 3 (b)), the optimized $\epsilon_{src}$ is used for getting the target-aware intermediate state $x_t^{src}$ and then to bootstrap a better editing result.

### 3.3.1 SOURCE TO TARGET

We first randomly sample a noise $\epsilon_{src}$ from the standard normal distribution and parameterize it as a learnable vector. Then, we follow the forward process of FM to add $\epsilon_{src}$ to the source image $x_0^{src}$:

$$x_t^{src} = (1-t) * x_0^{src} + t * \epsilon_{src} \quad t \in (0,1), \tag{8}$$

where $x_t^{src}$ is the intermediate state obtained by linearly interpolating the source image and noise. The interpolation ratio is controlled by $t$. We take $x_t^{src}$ as the starting point of the following restoration process, *i.e.*, $x_t^{tar} = x_t^{src}$. By inputting $x_t^{tar}$ into the pretrained FM model (*i.e.*, the velocity prediction network $v_\theta$) under the guidance of target prompt $c_{tar}$, we can get the predicted velocity $v_\theta(x_t^{tar}, t, c_{tar})$. Then we take the Euler solver to denoise it into the target image $x_0^{tar}$:

$$x_{t-\Delta}^{tar} = x_t^{tar} - v_\theta(x_t^{tar}, t, c_{tar}) * \Delta \Rightarrow x_0^{tar}, \tag{9}$$

where $\Delta$ is the Euler step size. After the above process, we can already get a naive target editing image $x_0^{tar}$ (*e.g.*, SDEdit (Meng et al., 2021) directly takes it as the final editing result). However, it turns out $x_0^{tar}$ suffers from the low *source consistency* because the random corruption indiscriminately destroyed the information of the source image.

### 3.3.2 TARGET TO SOURCE

To further improve $x_0^{tar}$, we try to recover the source image from $x_0^{tar}$. Considering its symmetry nature, a better recovery result indicates that $x_0^{tar}$ is of higher *source consistency*. Specifically, we parameterize another noise $\epsilon_{tar}$ (initialized from standard Gaussian noise) and add it to $x_0^{tar}$:

$$z_t^{tar} = (1-t) * x_0^{tar} + t * \epsilon_{tar} \quad t \in (0,1), \tag{10}$$

where $z_t^{tar}$ is the noised version (another intermediate state) of the generated target image $x_0^{tar}$. We take $z_t^{tar}$ as the starting point of the following restoration process, *i.e.*, $z_t^{src} = z_t^{tar}$ and denoise it under the guidance of source prompt $c_{src}$:

$$z_{t-\Delta}^{src} = z_t^{src} - v_\theta(z_t^{src}, t, c_{src}) * \Delta \Rightarrow z_0^{src}, \tag{11}$$

where $z_0^{src}$ is the recovery of source image. The better recovery (*i.e.*, $z_0^{src}$ is closer to $x_0^{src}$) indicates that the generated target image $x_0^{tar}$ maintains editing-irrelevant information of the source image better. After the above two steps, we can conduct the following cycle-consistent optimization.

### 3.3.3 CYCLE-CONSISTENT OPTIMIZATION

As shown in Figure 3 (a), the recovery image $z_0^{src}$ is quite different from the source image $x_0^{src}$ because the two intermediate states (*i.e.*, $x_t^{src}$ and $z_t^{tar}$) of the whole cycle were indiscriminately destroyed and the global source information was lost. To get the ideal target-aware intermediate state, we constrain the whole cycle to satisfy two cycle-consistent objectives:

1) The two intermediate states $x_t^{src}$ and $z_t^{tar}$ should share the aligned semantics as corrupted images, *i.e.*, destroy the editing-relevant parts of the source image (*e.g.*, `white dog` and `cat`) while maintaining the editing-irrelevant parts (*e.g.*, `background` and `car`). To achieve that, we use an alignment constraint to force $x_t^{src}$ and $z_t^{tar}$ to be close in the latent space:

$$\mathcal{L}_{align} = \|x_t^{src} - z_t^{tar}\|_2^2. \tag{12}$$

2) Starting from $z_t^{tar}$, the recovery result (*i.e.*, $z_0^{src}$) should be highly similar to the source image $x_0^{src}$ because the ideal $z_t^{tar}$ maintains the editing-irrelevant parts. Thus, we take a recovery constraint to encourage $z_t^{tar}$ to be a good start point of recovering to the source image $x_0^{src}$:

$$\mathcal{L}_{rec} = \|z_0^{src} - x_0^{src}\|_2^2. \tag{13}$$

To decrease the above two losses, we leverage the property that $\epsilon_{src}, \epsilon_{tar}$ can effectively rectify $x_t^{src}$ and $z_t^{tar}$ based on Eq. (8) and Eq. (10) and further rectify the whole cycle. Thus, we can jointly optimize the two learnable noises to decrease the $\mathcal{L}_{align}$ and $\mathcal{L}_{rec}$:

$$\epsilon_{src}, \epsilon_{tar} = \underset{\epsilon_{src}, \epsilon_{tar}}{\arg\min}(\mathcal{L}_{rec} + \lambda\mathcal{L}_{align}), \tag{14}$$

where $\lambda$ is a weight hyperparameter to balance the two loss items. Through the above cycle-consistent optimization, $\epsilon_{src}, \epsilon_{tar}$ can discriminately destroy the information according to whether it is editing-relevant. Then the intermediate state $x_t^{src}$ can be rectified towards the ideal role to be both editable and editing-irrelevant preserved. Thus, in the inference state, we can directly denoise the target-aware intermediate state $x_t^{src}$ to get an improved editing result (*c.f.*, Figure 3 (b)).

Table 1: **Quantitative comparisons with five baselines for image editing on PIEBench**. The best results are highlighted in bold, while the second-best results are underlined.

| Method | Source Consistency | | | | | Semantic Alignment | |
|--------|----------|--------|--------|-------|-------|--------------|--------------|
| | Distance ↓ | PSNR ↑ | LPIPS ↓ | MSE ↓ | SSIM ↑ | CLIP Entire ↑ | CLIP Edited ↑ |
| ODE-Inv | 0.074 | 17.57 | 0.240 | 0.024 | 0.691 | 24.57 | 21.73 |
| SDEdit (2021) | 0.036 | 22.57 | 0.119 | 0.008 | 0.747 | 24.56 | 21.95 |
| iRFDS (2024) | 0.069 | 18.81 | 0.191 | 0.021 | 0.738 | 25.12 | 21.95 |
| FlowEdit (2024) | 0.036 | 23.02 | 0.082 | 0.007 | 0.842 | **25.98** | **22.81** |
| FlowAlign (2025a) | 0.028 | 25.50 | **0.053** | 0.004 | 0.879 | 25.28 | 22.00 |
| **Ours** | **0.013** | **26.83** | 0.063 | **0.003** | **0.886** | 25.48 | 22.46 |

## 4 EXPERIMENTS

**Implementation Details**. We used SD-3-medium[2] (Esser et al., 2024) from HuggingFace as the pretrained text-to-image flow model. The overall timestep number is set to $50$ and the corruption step $t = 33$ (*c.f.*, Eq. (8)). During the optimization, the source guidance scale and target guidance scale were set to 3.5 and 5.5, respectively. We used the Adam optimizer (Kingma, 2014) with $1e-1$ learning rate for 100 steps of noise optimization. The loss weight $\lambda$ was set to 0.2. In the inference stage, the target guidance scale was 5.5. All experiments were conducted on an RTX 3090 GPU.

**Benchmark**. We evaluated our method on PIE-Bench (Ju et al., 2024), a text-based image editing benchmark. It contains 700 (source image, source prompt, target prompt) tuples, which cover nine editing types (*e.g.*, "add object" and "change style"). For the evaluation metrics, we reported the seven commonly used metrics to evaluate both source consistency (*Structure Distance*, *Background PSNR*, *Background LPIPS*, *Background MSE*, *Background SSIM*) and Semantic Alignment (*Entire CLIP Score* and *Edited CLIP Score*). More details are in the Appendix A.1.

**Baselines**. We compared our method with five state-of-the-art flow-based editing methods that were implemented with SD-3-medium: 1) *ODE inversion* (*ODE-Inv*), which gets the inversion by using the Euler Solver and denoises with the target prompt. 2) *SDEdit* (Meng et al., 2021), which adds random noise and then denoises with the target prompt. 3) *iRFDS* (Yang et al., 2024), which is a SDS-based editing method. 4) *FlowEdit* (Kulikov et al., 2024) and 5) *FlowAlign* (Kim et al., 2025a) are two inversion-free editing methods. The results and hyperparameters for the reimplementation of these methods follow previous works. More details are in the Appendix A.3.

### 4.1 COMPARISONS WITH STATE-OF-THE-ARTS

**Quantitative Evaluation**. As shown in Table 1, by only optimizing noises under the cycle constraints and getting a target-aware intermediate state, our method can maintain a remarkable source consistency (*i.e.*, on four of five metrics our method gets the best results and the left one is the second-best). The source consistency is a significant outcome benefited from the cycle consistency constraints. At the same time, our method can also achieve a good semantic alignment to the target prompt with the second-best CLIP Score regarding both the entire and the edited parts. This indicates that our method can achieve a good balance between source consistency and semantic alignment. We provide more quantitative results in the Appendix A.1.

**Qualitative Evaluation**. As shown in Figure 4, we gave qualitative comparisons with four baselines except FlowAlign (not open-sourced now). We have two observations: 1) Our method produced high-quality editing results (high source consistency and semantic alignment) on cases from various editing types. This indicates that our method is generally effective for extensive editing requirements. 2) As a highlight, our method can significantly maintain the editing-irrelevant parts. This advantage exactly verifies that the target-aware corruption can smartly hold the editing-irrelevant source content while making the editing-relevant contents editable.

### 4.2 ABLATION STUDY

**Optimization Steps Ablation**. To analyze how optimization with cycle constraints can benefit the editing performance, we provided two visualized examples in Figure 5. Each example con-

---

[2]Our method is also compatible with other models like DDPM-based SD-1.5, *c.f.*, Appendix A.5.

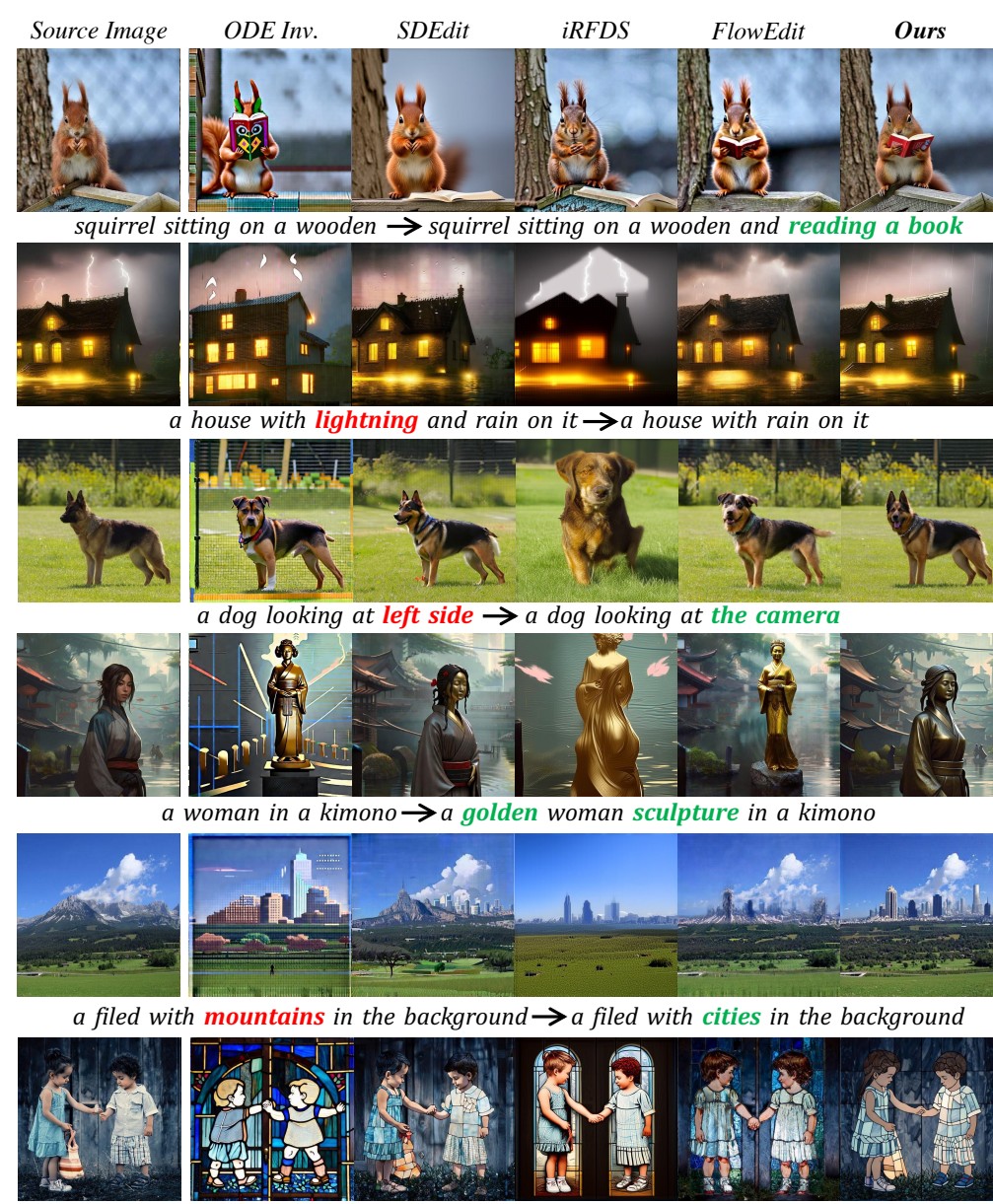

Figure 4: **Qualitative comparisons with four baselines**. The examples are across six editing types, including "add object", "delete object", "change pose", "change material", "change background" and "change style". More editing types are in the Appendix A.8.

tained the editing results of every 20 steps during the optimization. Before the optimization (*i.e.*, step 0), it already achieves a good target semantic alignment. For example, the `cat` has already appeared to replace the `rabbit` (step 0 of the first row), and the `girl's short hair` has also appeared to replace the `long hair` (step 0 of the second row). However, the editing-irrelevant parts (*e.g.*, `background`, `candies`, and `girl`) are quite different from the source images. As the optimization proceeds, the source consistency has gradually improved, while the target semantics remain well-aligned. This indicates that our cycle constraints can significantly maintain the editing-irrelevant parts while editing the target parts. More quantitative results are in the Appendix A.2.

**Generalization of the Intermediate State**. To further verify our main claim (*i.e.*, the target-aware intermediate state can maintain the editing-irrelevant contents and destroy the editing-relevant contents), we evaluated the generalization ability of the target-aware intermediate state. As shown in Figure 6, we take the 1-4 columns of the first row as the example. The source image is "velvet cake

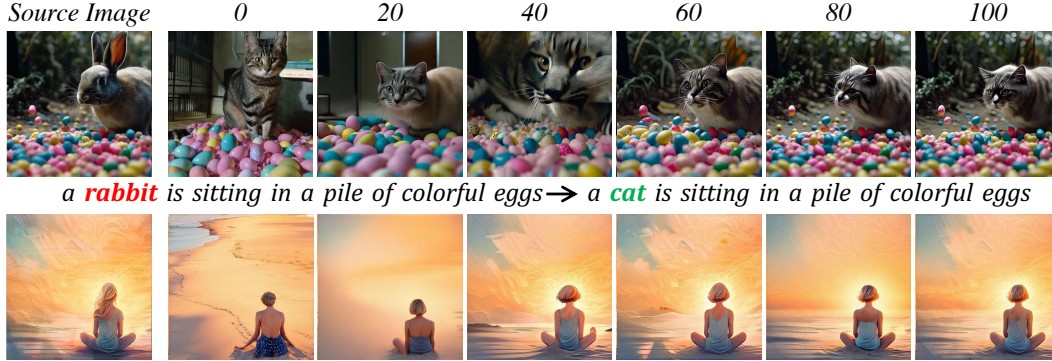

Figure 5: **Optimization Steps ablation**. From left to right, the quality of the editing result is gradually improved (better source consistency) across optimization steps.

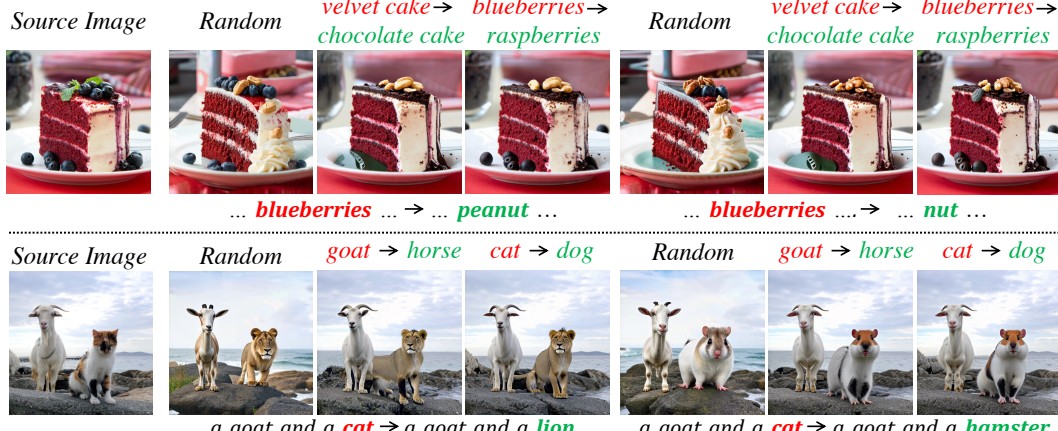

Figure 6: **Generalization ability of intermediate state**. The intermediate state is acquired by random corruption or optimizing with the target shown above the images (*e.g.*, `velvet cake` → `chocolate cake`). The transferred target is shown below the images.

with blueberries" and the editing target is "change blueberries to peanuts". We restored from three intermediate state (shown above the image): 1) Random State is acquired by adding random noise to the source image. 2) Different-pattern State is acquired by optimizing on another totally different editing target (*i.e.*, "change velvet cake to chocolate cake") with our method. 3) Similar-pattern State is acquired by optimizing on another similar editing target (*i.e.*, "change blueberries to raspberries"). We can find that restoration from a similar-pattern state leads to a better editing result compared to random and different patterns. This verifies our claim of the intermediate state and demonstrates some good generalization ability of target-aware intermediate states across similar editing patterns.

## 5 CONCLUSION AND DISCUSSION

In this paper, we revealed the limitations of the target-agnostic intermediate state in the existing corruption-then-restoration editing paradigm and proposed to acquire target-aware intermediate states by optimizing two noises with two cycle-consistent constraints. To satisfy the cycle-consistent constraints, the editing result is improved to maintain the editing-irrelevant contents of the source image for better source consistency. We conducted extensive quantitative, qualitative, and ablation experiments to show the effectiveness of our method. Moreover, we found that the target-aware intermediate state is transferable within similar editing patterns.

**Limitation**. Since our method is optimization-based and usually converges after a few steps, it is relatively time-consuming (*c.f.*, Appendix A.4) compared with other optimization-free methods. In the future, we are going to: 1) Explore a more efficient optimization mechanism. 2) Instead of optimization, directly acquire the target-aware intermediate state that can satisfy the Cycle Consistency.

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

# A APPENDIX

This appendix is organized as follows:

- Section A.1 provides additional quantitative results of our method on PIE-Bench. We reported 18 evaluation metrics across 11 groups for more comprehensive evaluations.
- Section A.2 gives the quantitative results of the ablations for optimization steps (*c.f.*, Sec. 4 Optimization Steps Ablation).
- Section A.3 indicates the implementation details of competing methods, including how we get the quantitative and qualitative results of them.
- Section A.4 gives more ablation results for the hyperparameters and analyzes the computational overhead of our method.
- Section A.5 shows that our method is compatible with DDPM-based models, *e.g.*, SD-1.5.
- Section A.6 provides the justifications for why FlowEdit can be classified as Injection-based Restoration.
- Section A.7 gives the implementation details of how we get the visualization of Figure 1 as well as more examples.
- Section A.8 provides additional qualitative results, including comparison with baselines and some editing examples of our method.

## A.1 ADDITIONAL QUANTITATIVE RESULTS

PIE-Bench (Ju et al., 2024) is a comprehensive benchmark for the text-based image editing task. It contains 700 images on nine editing types: "change object", "add object", "delete object", "change attribute content", "change attribute pose", "change attribute color", "change attribute material", "change background", and "change style". Moreover, it contains 18 evaluation metrics: *PSNR*, *LPIPS* (Zhang et al., 2018), *MSE*, *SSIM* (Wang et al., 2004), *Structure Distance* (Tumanyan et al., 2022), *Edited PSNR*, *Edited LPIPS*, *Edited MSE*, *Edited SSIM*, *Edited Structure Distance*, *Background PSNR*, *Background LPIPS*, *Background MSE*, *Background SSIM*, *Background Structure Distance*, *CLIP Score to source image* (Wu et al., 2021), *Entire CLIP Score*, *Edited CLIP Score*. More benchmark details, including the metrics explanation and datasets statistics, can be found in Ju et al. (2024). Table 2 and Table 3 give the results of 18 evaluation metrics across 11 groups (9 editing types, a random group, and the overall group).

## A.2 QUANTITATIVE RESULTS OF THE ABLATIONS FOR OPTIMIZATION

In the Optimization Steps Ablation of Sec. 4, we provide the qualitative results across optimization steps on the whole PIE-Bench. Here we give the detailed quantitative results in Figure 7. We can see that the *Structure Distance* (reflecting the source consistency) is significantly decreased with the optimization, while a little decrease in the *Edited CLIP Score* (reflecting the target semantic alignment). This indicates that our method can achieve a good balance between the source consistency and target alignment.

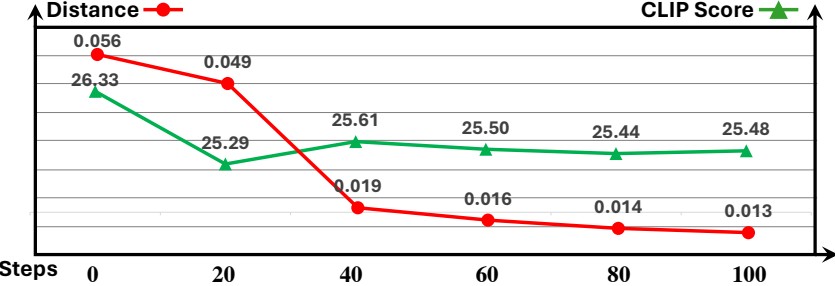

Figure 7: The curve of the entire Structure Distance and the Edited CLIP Score across the optimization steps.

Table 2: The quantitative results of our method across 11 groups (random group, nine editing types, and overall group). Metrics of the entire image and the Background part are reported.

| Editing Types | Entire Image | | | | | Background Part | | | | |
|---|---|---|---|---|---|---|---|---|---|---|
| | PSNR | LPIPS | MSE | SSIM | Distance | PSNR | LPIPS | MSE | SSIM | Distance |
| Random | 22.46 | 0.127 | 0.007 | 0.789 | 0.014 | 27.91 | 0.058 | 0.003 | 0.894 | 0.011 |
| Change Object | 21.13 | 0.145 | 0.010 | 0.769 | 0.019 | 25.19 | 0.078 | 0.004 | 0.866 | 0.009 |
| Add Object | 23.59 | 0.106 | 0.005 | 0.832 | 0.010 | 27.22 | 0.057 | 0.003 | 0.904 | 0.005 |
| Delete Object | 22.87 | 0.111 | 0.006 | 0.808 | 0.010 | 25.04 | 0.077 | 0.004 | 0.856 | 0.006 |
| Change Content | 23.61 | 0.108 | 0.005 | 0.806 | 0.012 | 27.34 | 0.063 | 0.002 | 0.896 | 0.007 |
| Change Pose | 23.66 | 0.095 | 0.005 | 0.814 | 0.015 | 27.14 | 0.065 | 0.003 | 0.868 | 0.012 |
| Change Color | 23.93 | 0.100 | 0.006 | 0.813 | 0.011 | 26.98 | 0.063 | 0.003 | 0.882 | 0.011 |
| Change Material | 22.77 | 0.114 | 0.007 | 0.807 | 0.012 | 29.48 | 0.055 | 0.002 | 0.914 | 0.004 |
| Change Background | 23.10 | 0.127 | 0.006 | 0.797 | 0.013 | 26.56 | 0.045 | 0.003 | 0.907 | 0.003 |
| Change Style | 22.11 | 0.148 | 0.007 | 0.772 | 0.014 | 28.04 | 0.066 | 0.004 | 0.890 | 0.003 |
| Overall | 22.75 | 0.122 | 0.007 | 0.798 | 0.013 | 26.83 | 0.063 | 0.003 | 0.886 | 0.008 |

Table 3: The quantitative results of our method across 11 groups (random group, nine editing types, and overall group). Metrics of the edited part and CLIP Scores are reported.

| Editing Types | Edited Part | | | | | CLIP Score | | |
|---|---|---|---|---|---|---|---|---|
| | PSNR | LPIPS | MSE | SSIM | Distance | Source | Entire | Edited |
| Random | 25.25 | 0.068 | 0.005 | 0.885 | 0.006 | 25.68 | 25.16 | 22.75 |
| Change Object | 25.15 | 0.057 | 0.006 | 0.909 | 0.006 | 26.24 | 25.04 | 21.01 |
| Add Object | 26.44 | 0.064 | 0.003 | 0.898 | 0.006 | 25.70 | 26.41 | 24.37 |
| Delete Object | 28.87 | 0.024 | 0.002 | 0.957 | 0.002 | 27.14 | 24.10 | 17.63 |
| Change Content | 27.25 | 0.045 | 0.003 | 0.903 | 0.008 | 25.78 | 25.16 | 21.98 |
| Change pose | 28.21 | 0.033 | 0.002 | 0.933 | 0.006 | 26.94 | 26.33 | 22.00 |
| Change Color | 28.06 | 0.033 | 0.003 | 0.932 | 0.004 | 26.77 | 25.58 | 20.61 |
| Change Material | 25.14 | 0.049 | 0.004 | 0.900 | 0.004 | 25.87 | 25.73 | 23.94 |
| Change Background | 26.85 | 0.082 | 0.004 | 0.880 | 0.008 | 25.85 | 24.90 | 22.49 |
| Change Style | 22.17 | 0.146 | 0.007 | 0.775 | 0.014 | 25.59 | 27.07 | 26.93 |
| Overall | 26.05 | 0.065 | 0.004 | 0.892 | 0.007 | 26.07 | 25.48 | 22.46 |

## A.3 IMPLEMENTATION DETAILS OF COMPETING METHODS

In Section. 4, we compared our method with five competing methods: 1) *ODE inversion* (*ODE-Inv*). 2) *SDEdit* (Meng et al., 2021). 3) *iRFDS* (Yang et al., 2024). 4) *FlowEdit* (Kulikov et al., 2024). 5) *FlowAlign* (Kim et al., 2025a). Here we provide the implementation details of these five methods:

- *ODE-Inv*: We followed the setting of Kulikov et al. (2024), *i.e.*, the overall timestep number is 50, the inversion step was set to 33. The source guidance scale and target guidance scale were set to 3.5 and 13.5, respectively.

- *SDEdit*: We followed the setting of Kim et al. (2025a), *i.e.*, the overall timestep number is 50, the corruption step $t$ was set to 18. The target guidance scale was set to 13.5.

- *iRFDS*: We followed the official implementations [3].

- *FlowEdit*: We followed the setting of Kim et al. (2025a), *i.e.*, the overall timestep number is 50, the corruption step $t$ was set to 33. The source guidance scale and target guidance scale were set to 3.0 and 13.5, respectively.

- *FlowAlign*: Since it is not open-source now, we directly reported the results from the paper.

For the qualitative results of four competing methods (except *FlowAlign*), we followed the above implementations. For the quantitative results of *SDEdit*, *FlowEdit*, and *FlowAlign*, we directly reported the results from Kim et al. (2025a). For the quantitative results of *ODE-Inv* and *iRFDS*, we followed the above implementations.

## A.4 HYPER-PARAMETERS ABLATIONS AND COMPUTATIONAL OVERHEAD ANALYSIS

**Hyper-parameters Ablations**. In our method, there are three important hyperparameters: 1) The source guidance scale (Source CFG). 2) The target guidance scale (Target CFG). 3) The loss bal-

---

[3] https://github.com/yangxiaofeng/rectified_flow_prior

Table 4: The ablations of three key hyperparameters of our method. We adopted the best-performing setting as our default (in gray).

| Source CFG | Target CFG | $\lambda$ | Distance | PSNR | LPIPS | MSE | SSIM | CLIP Entire | CLIP Edited |
|---|---|---|---|---|---|---|---|---|---|
| 3.5 | 5.5 | 0.05 | 0.012 | 27.12 | 0.061 | 0.003 | 0.888 | 25.23 | 22.10 |
| 3.5 | 5.5 | 0.2 | 0.013 | 26.83 | 0.063 | 0.003 | 0.886 | 25.48 | 22.46 |
| 3.5 | 5.5 | 0.5 | 0.015 | 26.35 | 0.069 | 0.004 | 0.877 | 25.55 | 22.52 |
| 3.5 | 5.5 | 0.2 | 0.013 | 26.83 | 0.063 | 0.003 | 0.886 | 25.48 | 22.46 |
| 5.5 | 5.5 | 0.2 | 0.014 | 26.58 | 0.068 | 0.003 | 0.880 | 25.14 | 22.00 |
| 2.5 | 5.5 | 0.2 | 0.014 | 26.41 | 0.066 | 0.003 | 0.880 | 25.92 | 22.26 |
| 2.5 | 2.5 | 0.2 | 0.019 | 24.88 | 0.085 | 0.005 | 0.861 | 26.09 | 22.99 |

Table 5: The time cost and GPU memory cost comparisons across two diffusion models and two GPU types (RTX 3090 and 80G-A800).

| Setting | | 3090 (s) | A800 (s) | GPU Memeroy (GB) |
|---|---|---|---|---|
| SD1.5 | Null-text | 207 | 119 | 11 |
| | Ours | 230 | 137 | 6 |
| SD3-medium | iRFDS | - | 132 | 34 |
| | Ours | 596 | 247 | 16 |

ance weight $\lambda$ (*c.f.*, Sec. 3.3.3). To evaluate how these hyperparameters influence the editing performance, we ablated them on PIE-Bench in Table 4. From the results of rows 1-3, we can see that a larger weight for the $\mathcal{L}_{align}$ (*i.e.*, a larger $\lambda$) leads to a lower source consistency and a higher target alignment. From rows 4-7, we can see that the balance between Source CFG and Target CFG can influence the balance between source consistency and target alignment.

**Computational Overhead Analysis**. As we mentioned in the Limitation part, since our method is an optimization-based method, the speed of editing is relatively low compared to the optimization-free methods. Here, we provided the computational cost comparisons with another two optimization editing methods: Null-text inversion (Mokady et al., 2023) (*Null-text*) and *iRFDS* (Yang et al., 2024) in Table 5. We can see that our method has a similar order of magnitude of time overhead to other methods, with a lower GPU memory cost. In the future, we can conduct research on how to increase the time efficiency of our method from both the algorithm perspective and from the hardware perspective.

## A.5 APPLICABILITY TO DDPM-BASED MODELS

Since our method is based on optimizing noises to get a target-aware intermediate state with the cycle-consistent constraints, it's inherently applicable to general diffusion models with different forward SDEs (Song et al., 2020b) like DDPM-based SD-1.5 (Rombach et al., 2022).

**Settings**. To verify the applicability of our method to DDPM-based models, we replace the flow matching model SD-3-medium in Sec. 4 with the DDPM-based SD-1.5. And compare our method with four DDPM-based baselines. 1) *SDEdit* (Meng et al., 2021): the overall timestep number is 25, the corruption step $t$ was set to 18. The target guidance scale was set to 5.5. 2) DDIM inversion (*DDIM*). 3) DDIM inversion + Prompt-to-Prompt (Hertz et al., 2022) (*DDIM + P2P*). 4) DDIM inversion + pix2pix-zero (Parmar et al., 2023) (*DDIM + P2P-Zero*). For *DDIM*, we followed the setting and reported the results from Mao et al. (2025). For *DDIM + P2P* and *DDIM + P2P-Zero*, we followed the setting and reported the results from Ju et al. (2024). For our method, the overall timestep number is set to 25 and the corruption step $t = 18$ (*c.f.*, Eq. (8)). During the optimization, the source guidance scale and target guidance scale were set to 3.5 and 5.5, respectively. We used the Adam optimizer (Kingma, 2014) with $1e - 2$ learning rate for 50 steps of noise optimization. The loss weight $\lambda$ was set to 0.5. In the inference stage, the target guidance scale was 5.5. All experiments were conducted on an RTX 3090 GPU.

**Results**. As shown in Table 6, our method can achieve the best balance between source consistency and semantic alignment. This indicates that our noise optimization under cycle constraints is generally effective for both the flow-based model and the DDPM-based model. Besides, we also reported the time and memory cost in Table 5. Finally, we gave some editing examples of our method on SD-1.5 in Figure 8.

Table 6: **Quantitative comparisons with four baselines of DDPM-based models for image editing**. The best results are highlighted in bold, while the second-best results are underlined.

| Method | Source Consistency | | | | | Semantic Alignment | |
|---|---|---|---|---|---|---|---|
| | Distance ↓ | PSNR ↑ | LPIPS ↓ | MSE ↓ | SSIM ↑ | CLIP Entire ↑ | CLIP Edited ↑ |
| SDEdit | 0.049 | 19.81 | 0.193 | 0.015 | 0.709 | 26.00 | 22.97 |
| DDIM | 0.080 | 17.36 | 0.220 | 0.024 | 0.705 | **27.08** | **23.90** |
| DDIM + P2P | 0.070 | 17.87 | 0.209 | 0.022 | 0.716 | 25.28 | 22.57 |
| DDIM + P2P-Zero | 0.062 | 20.44 | 0.172 | 0.014 | **0.747** | 22.80 | 22.54 |
| **Ours** | **0.040** | **20.81** | **0.169** | **0.013** | 0.725 | 26.02 | 22.95 |

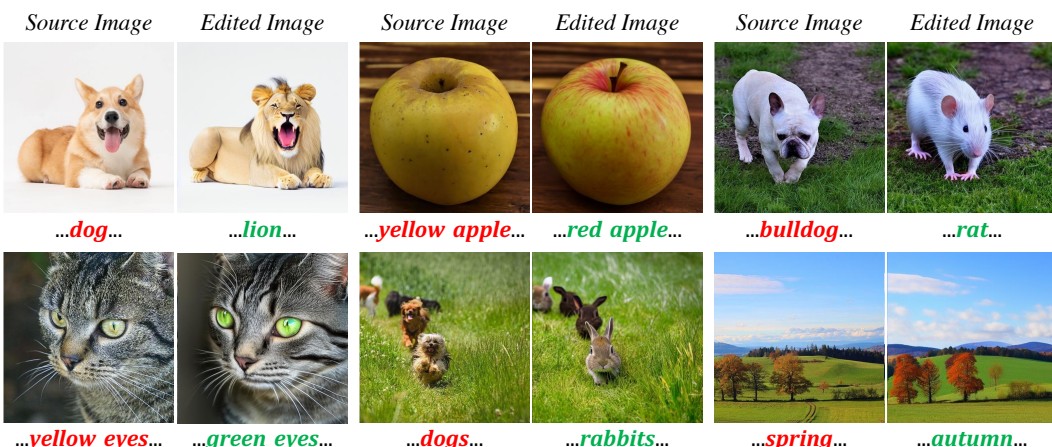

*Source Image*  *Edited Image*  *Source Image*  *Edited Image*  *Source Image*  *Edited Image*

...*dog*...  ...*lion*...  ...*yellow apple*...  ...*red apple*...  ...*bulldog*...  ...*rat*...

...*yellow eyes*...  ...*green eyes*...  ...*dogs*...  ...*rabbits*...  ...*spring*...  ...*autumn*...

Figure 8: The quantitative examples of our method on SD-1.5.

### A.6 JUSTIFICATION FOR CLASSIFYING FLOWEDIT AS INJECTION-BASED RESTORATION

---
**Algorithm 1** Algorithm for FlowEdit

---
1: **Input:** real image $X^{src}$, $\{t_i\}_{i=0}^{T}$, $n_{max}$
2: **Output:** edited image $X^{tar}$
3: **Init:** $Z_{t_{max}}^{FE} = X_0^{src}$
4: **for** $i = n_{max}$ **to** 1 **do**
5:     $N_{t_i} \sim \mathcal{N}(0,1)$
6:     $Z_{t_i}^{src} \leftarrow (1-t_i)X^{src} + t_i N_{t_i}$
7:     $Z_{t_i}^{tar} \leftarrow Z_{t_i}^{FE} + Z_{t_i}^{src} - X^{src}$
8:     $V_{t_i}^{\Delta} \leftarrow V^{tar}(Z_{t_i}^{tar}, t_i) - V^{src}(Z_{t_i}^{src}, t_i)$
9:     $Z_{t_{i-1}}^{FE} \leftarrow Z_{t_i}^{FE} + (t_{i-1}-t_i)V_{t_i}^{\Delta}$
10: **end for**
11: **Return:** $Z_0^{FE} = X_0^{tar}$

---

We summarized current text-based editing methods into the correction-then-restoration paradigm. It is obvious to claim some methods to explicitly inject source condition during the restoration stage (*e.g.*, injecting the attention maps). However, for some methods, they implicitly inject the source condition, such as FlowEdit (Kulikov et al., 2024). Thus, we provided the justification here. The pseudo code of FlowEdit is shown in Algorithm 1. Consider Line 9:

$$Z_{t_{i-1}}^{FE} \leftarrow Z_{t_i}^{FE} + (t_{i-1}-t_i)V_{t_i}^{\Delta}. \tag{15}$$

We can substitute the $V_{t_i}^{\Delta}$ from Line 8 into it and get:

$$Z_{t_{i-1}}^{FE} \leftarrow Z_{t_i}^{FE} + (t_{i-1}-t_i)(V^{tar}(Z_{t_i}^{tar}, t_i) - V^{src}(Z_{t_i}^{src}, t_i)). \tag{16}$$

Simplified it to get:

$$Z_{t_{i-1}}^{FE} \leftarrow Z_{t_i}^{FE} + (t_{i-1}-t_i)V^{tar}(Z_{t_i}^{tar}, t_i) - (t_{i-1}-t_i)V^{src}(Z_{t_i}^{src}, t_i)). \tag{17}$$

Then, we can define an injection drift $\Delta_t$:

$$\Delta_t := -(t_{i-1}-t_i)V^{src}(Z_{t_i}^{src}, t_i)), \tag{18}$$

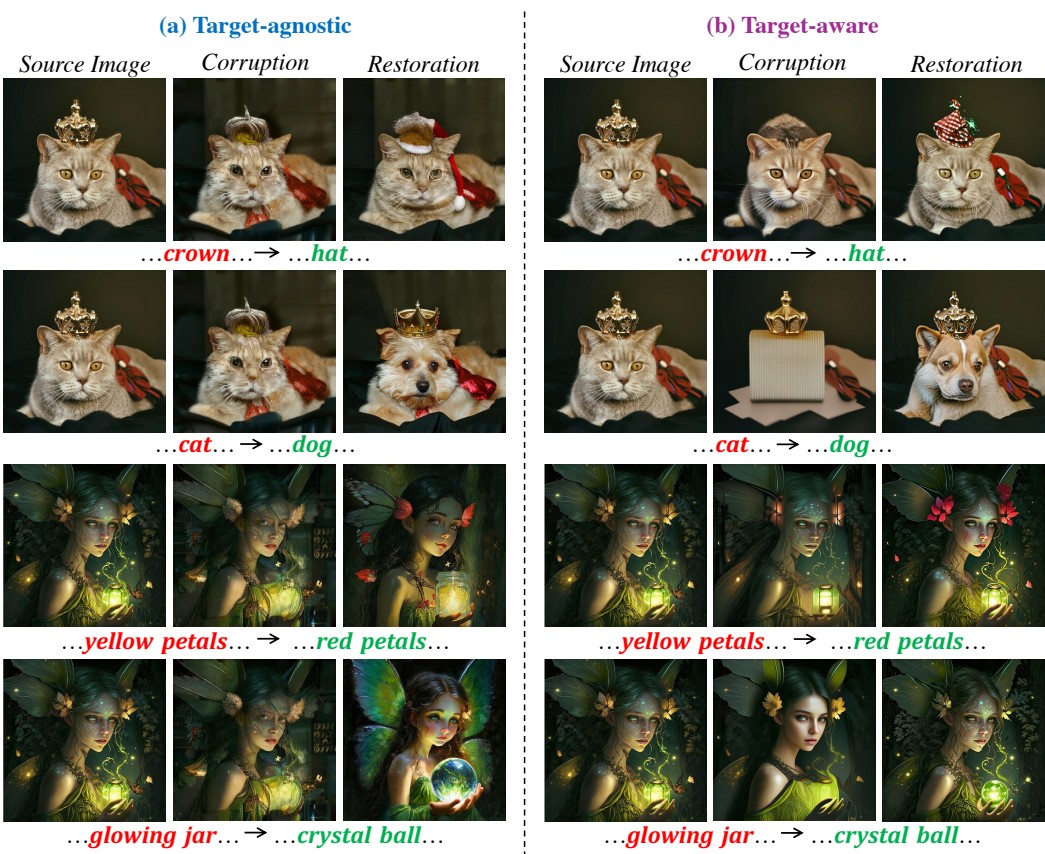

Figure 9: More cases to show the differences between Target-agnostic and Target-aware manners.

where this injection drift is only related to the source image. Substitute $\Delta_t$ into Eq. (17):

$$Z_{t_{i-1}}^{FE} \leftarrow Z_{t_i}^{FE} + (t_{i-1} - t_i)V^{tar}(Z_{t_i}^{tar}, t_i) + \Delta_t. \tag{19}$$

If we remove the $\Delta_t$ in Eq. (19), this restoration process degraded into a naive denoising process under the guidance of the target prompt. The only difference is that it adds an injection drift $\Delta_t$ (only related to the source image) for each update step. In that case, FlowEdit also follows the corruption-then-restoration paradigm and injects some source condition during the restoration stage, which is classified as inversion-free corruption and injection-based restoration (*c.f.*, Figure 2 (c)).

### A.7  IMPLEMENTATION DETAILS OF VISUALIZATION IN FIGURE 1

For the Target-agnostic manner, we adopted the SDEdit method as a representative, *i.e.* the intermediate state is obtained from randomly adding noise. Specifically, the overall timestep number was set to $50$, and the corruption step $t$ was set to $25$. The visualization of the corruption image was denoised from the intermediate state under the null text prompt (*i.e.*, ""). The visualization of the restoration image was denoised from the intermediate state under the target prompt with a 5.5 target guidance scale. For the Target-aware manner, we leveraged our method to get the target-aware intermediate state with a 3.5 source guidance scale and a 3.5 target guidance scale. The source prompt and target prompt only contained the different words (*e.g.*, source prompt was "cat" and target prompt was "tiger"). The visualization of the corruption image is also denoised from the intermediate state under the null text prompt (*i.e.*, ""). The final restoration result was followed by our default setting (*c.f.*, Sec. 4). We provide more examples in Figure 9.

### A.8  ADDITIONAL QUALITATIVE RESULTS

We provide additional qualitative comparisons with competing methods in Figure 10 and more editing examples of our FlowCycle in Figure 11.

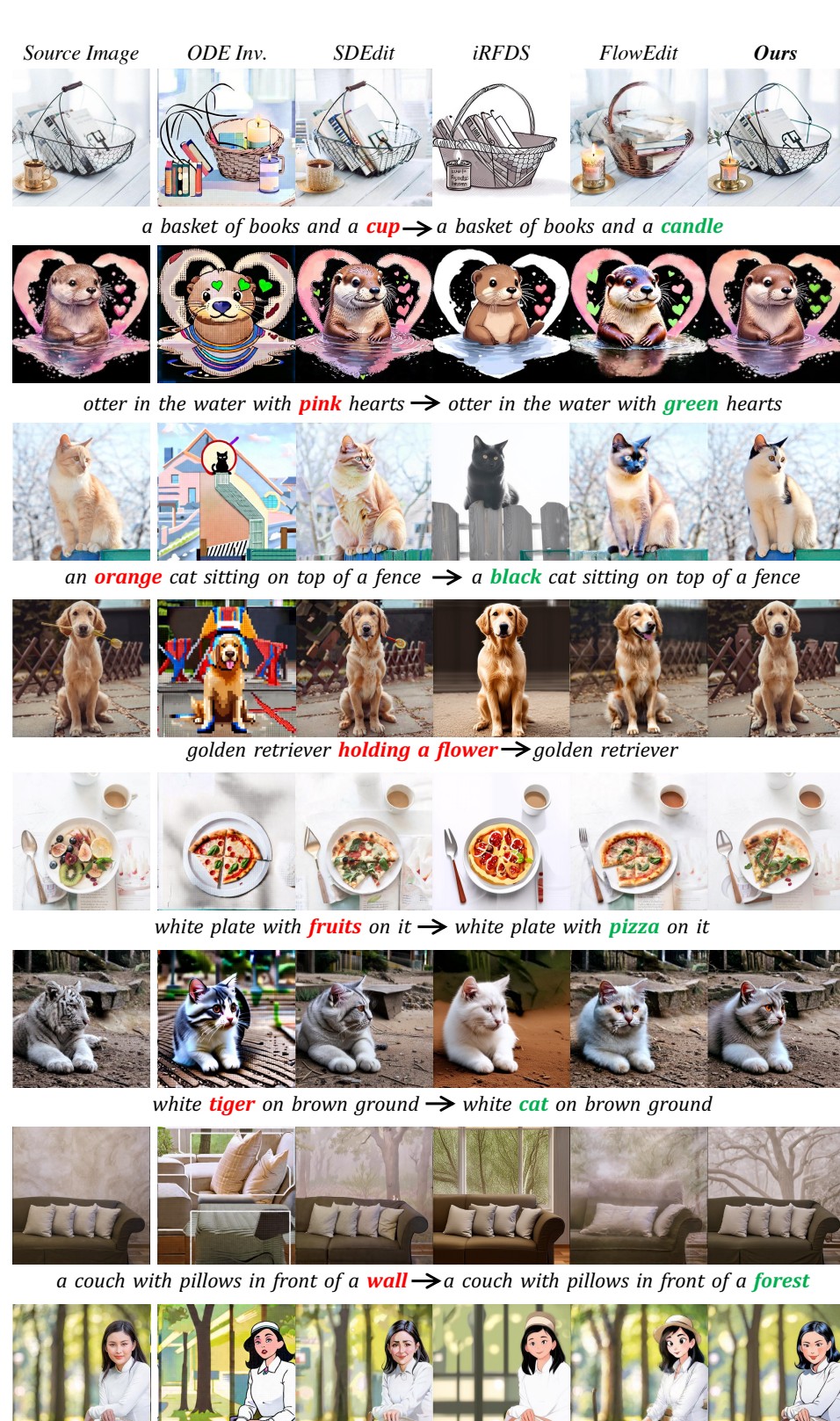

Figure 10: More quantitative comparisons with competing methods.

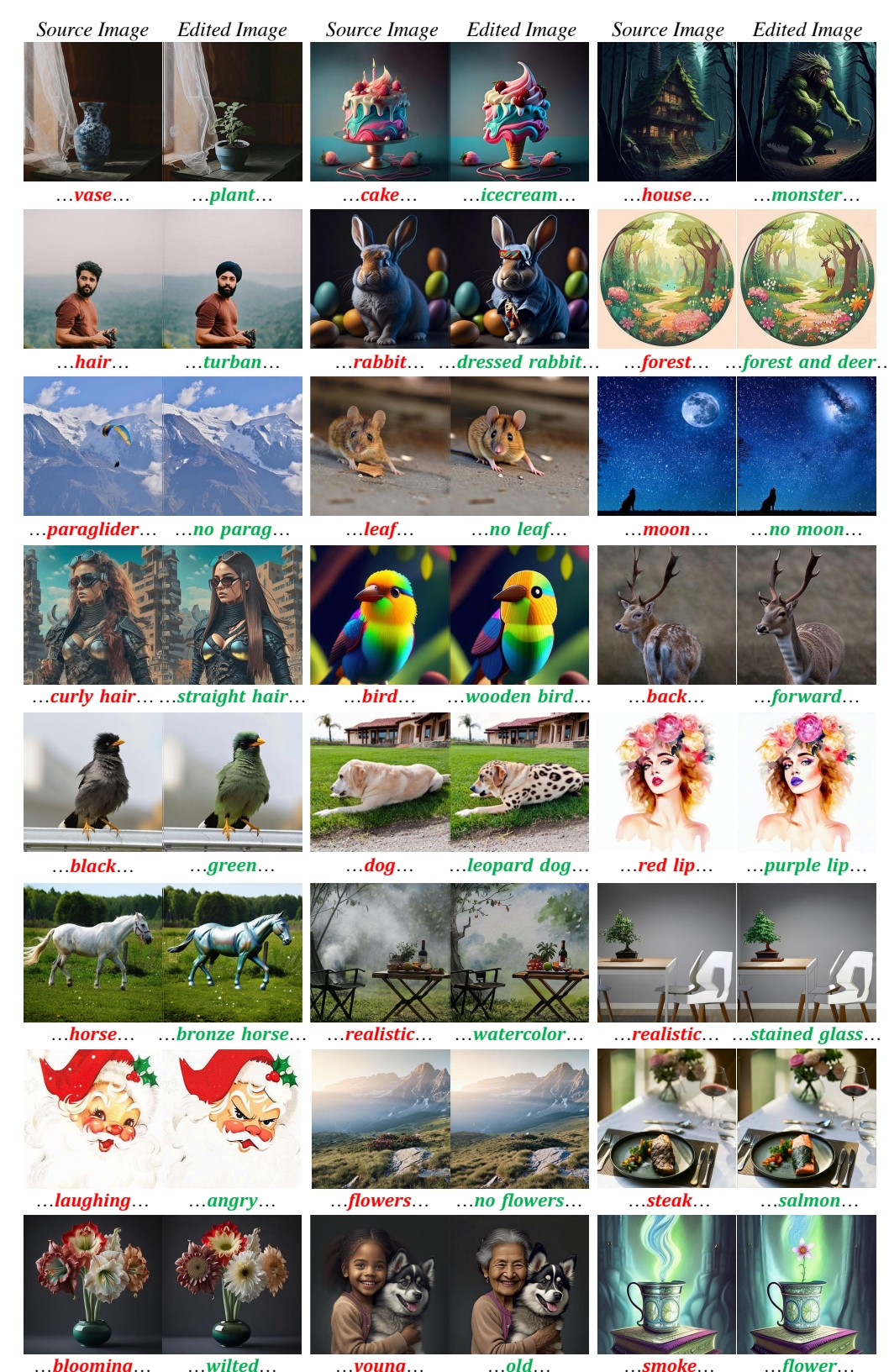

Figure 11: Additional quantitative examples of our FlowCycle.

