# OpenReview forum: "FlowCycle: Pursuing Cycle-Consistent Flows for Text-based Editing"
_ICLR.cc/2026/Conference — ICLR 2026 Conference Withdrawn Submission_

### Official Review · Reviewer_dhTm · 2025-10-20

**Soundness:** 3
**Presentation:** 3
**Contribution:** 2
**Rating:** 4
**Confidence:** 4

**Summary:**

The paper addresses text-based image editing with pre-trained rectified-flow models and argues that the usual corruption-then-restoration pipeline constructs a target-agnostic intermediate state, which limits editability and/or source consistency when the target edit departs from the source image. The authors propose FlowCycle, an inversion-free framework that learns a target-aware intermediate state by optimizing two learnable noises (ϵ_src, ϵ_tar) in a cycle between “source→target” and “target→source.” The optimization minimizes a recovery loss aligning the recovered image with the source and an alignment loss bringing the two corrupted intermediate states together, thereby encouraging corruption on editing-relevant regions while preserving editing-irrelevant content. Experiments on PIE-Bench using SD-3-medium report improved source-consistency metrics and competitive CLIP alignment versus recent baselines; ablations, qualitative results, and limited analysis on SD-1.5 are also provided.

**Strengths:**

* **Method quality and simplicity.** The mechanism—parameterizing corruption with learnable noises and optimizing with two MSE losses (recovery and alignment)—is simple to implement atop standard FM inference (Euler solver) and requires no architectural changes to the backbone. The losses and inference path are precisely specified.
* **Clarity.** The paper clearly situates FlowCycle within the corruption-then-restoration paradigm and articulates the three-step cycle (source→target, target→source, and cycle optimization), aided by a succinct objective and inference description.
* **Empirical significance.** On PIE-Bench, FlowCycle achieves the best or second-best results on most source-consistency metrics while maintaining solid CLIP alignment, indicating a favorable balance. Extensive qualitative comparisons and an optimization-steps ablation further support the claims.

**Weaknesses:**

* **Compute/latency overhead.** The approach is optimization-based; the authors acknowledge it is relatively time-consuming, and tabled timings show noticeable overhead versus optimization-free methods—raising practical concerns for interactive editing.
* **Per-edit optimization and generalization.** The learnable noises are optimized per source–target pair, with limited evidence on reuse. A figure suggests some transfer across similar edit patterns, but systematic protocols for reusing noises are not established.
* **Evaluation scope.** Experiments rely primarily on PIE-Bench with automatic metrics; there is no human study, no multi-turn editing evaluation, and limited testing beyond SD-3-medium (one SD-1.5 table), limiting claims of generality.
* **Comparative fairness and coverage.** Some baselines (e.g., FlowAlign) are reported from the paper rather than reproduced (not open-sourced), and it is unclear whether stronger recent editing techniques outside the flow family were considered.

**Questions:**

* **Sensitivity to hyperparameters.** How sensitive is performance to t, λ, and CFG scales? The appendix gives a table, but a more principled analysis (or defaults robust across datasets) would help practitioners.
* **Failure modes.** Please characterize typical failures (e.g., strong geometric edits, cluttered scenes, small objects) and whether L_align in pixel space encourages undesirable content averaging; are feature-space variants of L_align beneficial?
* **Beyond PIE-Bench.** Any plans to evaluate additional editing benchmarks or to include a small user study to corroborate automatic metrics, especially for perceived source consistency?

---

### Official Review · Reviewer_1QJ7 · 2025-10-27

**Soundness:** 2
**Presentation:** 2
**Contribution:** 2
**Rating:** 2
**Confidence:** 4

**Summary:**

FlowCycle is a flow-based, inversion-free framework that learns target-aware image editing through a cycle-consistent optimization process. It adds learnable noises to the source and target images, transforming them back and forth to enforce consistency between the two domains. By minimizing reconstruction and corruption losses, the model learns to focus corruption on editing-relevant regions while preserving other content. At inference time, the optimized noise produces a target-aware intermediate state, enabling accurate and consistent text-based image editing.

**Strengths:**

1.Unlike conventional approaches, FlowCycle selectively corrupts only editing-relevant regions, preserving unrelated content and achieving faithful, precise modifications.

2.By enforcing dual consistency between source and target images, it maintains structural and semantic integrity, ensuring strong source consistency even after major edits.

3.FlowCycle demonstrated superior performance across most evaluation metrics.

**Weaknesses:**

1.There is a lack of detailed comparison with baselines. As the authors explained in the Related Work section, existing text-based image editing methods encompass a variety of approaches. However, the authors only presented superior metric evaluations on PIEBench as evidence of performance.

2.Although flow-based methods are still in their early stages — and thus could potentially demonstrate large performance improvements — FlowCycle does not show a notably significant improvement compared to existing approaches.

3.While the method improves performance through a simple mechanism that incorporates MSE loss, there is some doubt as to whether this provides sufficient novelty or contribution.

**Questions:**

1.How does the time consumption of FlowCycle compare to inversion-free corruption methods? For instance, I think FlowAlign, which does not compute CFG for both the source and target, would likely show better efficiency in terms of time consumption.

2.FlowCycle shows very similar performance to FlowAlign across most metrics except for Distance. Please explain in more detail what advantages FlowCycle has over FlowAlign. The explanation provided in Section 4.1 of the paper seems somewhat insufficient.

3.The attempt to optimize FlowCycle is interesting; however, the use of MSE and λ appears to be a rather simple approach. For example, in the ablation study on λ presented in Table 4, the results seem more sensitive to changes in source CFG and target CFG than to λ itself. Is adding λ as a new parameter to the existing source and target CFG hyperparameters truly effective? What would happen if λ were set to 1 to give equal weight to alignment and reconstruction?

---

### Official Review · Reviewer_ZYcS · 2025-10-31

**Soundness:** 3
**Presentation:** 3
**Contribution:** 3
**Rating:** 6
**Confidence:** 4

**Summary:**

The paper introduces FlowCycle, a novel inversion-free, flow-based image editing framework that addresses a key limitation of existing text-based image editing methods: the target-agnostic nature of intermediate states. FlowCycle proposes a target-aware corruption strategy by parameterizing the corruption process with learnable noises optimized under cycle-consistency constraints. Extensive experiments on PIE-Bench show that FlowCycle achieves better source consistency and competitive semantic alignment compared to five state-of-the-art baselines.

**Strengths:**

- Novel conceptual insight: Identifies and addresses the overlooked issue of target-agnostic corruption in flow-based editing.
- Generalization evidence: Demonstrates that learned intermediate states generalize to similar editing patterns.

**Weaknesses:**

- Optimization cost: The method requires iterative noise optimization (≈100 steps), making it slower than optimization-free baselines. The authors acknowledge this but do not propose clear mitigation strategies.
- Limited theoretical depth: The paper motivates target awareness intuitively but lacks a rigorous theoretical link between the loss formulation and selective corruption behavior.
- Marginal semantic gains: While source consistency improves significantly, the improvement in semantic alignment (CLIP Edited score) over FlowAlign is relatively small.

**Questions:**

- How sensitive is the method to the λ parameter and timestep t — do small deviations significantly affect quality?
- Would integrating attention-based masks for editing-relevant regions further enhance target awareness?

---

### Official Review · Reviewer_LEif · 2025-11-04

**Soundness:** 3
**Presentation:** 3
**Contribution:** 3
**Rating:** 6
**Confidence:** 4

**Summary:**

FlowCycle is a text-based image editing framework for rectified-flow models that rethinks how the noisy intermediate state is constructed during editing. Instead of using target-agnostic noise or ODE inversion optimized only for reconstruction, the method learns two prompt-dependent noise tensors so that the corrupted state becomes more “target-aware,” disrupting editing-relevant regions while preserving the rest. These noises are optimized via a bidirectional cycle: source→target→source, with pixel-wise alignment at the intermediate step and reconstruction at the image level, while keeping the pretrained flow model frozen. Experiments on PIE-Bench and additional DDPM-based tests show markedly better source consistency and also competitive target alignment compared with recent flow-based editors such as FlowEdit and FlowAlign.

**Strengths:**

- The paper presents a concrete limitation of existing flow-based editors—target-agnostic intermediate states tuned for reconstruction rather than editing, then builds the entire method around overcoming this bottleneck.

- FlowCycle only optimizes noise tensors and does not fine-tune the underlying rectified-flow model, which makes it conceptually lightweight and easy to plug into a variety of backbones and even DDPMs.

- On PIE-Bench, the method achieves the best/near-best scores on most consistency metrics (e.g. bg quality, structural distance) while maintaining CLIP-based alignment close to the strongest baselines.

- The analysis of optimization steps, guidance scales, and the SD-1.5/DDPM variant shows that the core idea is not tied to a specific rectified-flow model.

- The paper situates FlowCycle among inversion-based, inversion-free, injection-based, and flow-based editing methods, making it clear how this work complements trajectory-regularization approaches and optimization-free designs.

**Weaknesses:**

- Each edit requires many optimization iterations and two denoising trajectories per iteration, which leads to latency and substantial GPU usage, which makes it hard to deploy in interactive editing tools.

- The method is motivated by selectively corrupting editing-relevant regions, yet the losses act uniformly in pixel space and the experiments do not report region-wise analysis of the intermediate state.

- Large structural changes, multi-object scenarios, or multi-attribute edits are not explored in depth, so the robustness of the approach in more demanding settings remains unclear.

- While the SD-1.5 experiments include some DDIM/SDEdit baselines, there is no detailed comparison to SOTA inversion-based diffusion editors under comparable conditions, making it harder to assess whether flow-based editing + FlowCycle is truly preferable in practice.

**Questions:**

- How sensitive is the method to the balance between intermediate alignment and source reconstruction losses, and have you explored schedules or adaptive weighting that might better handle large semantic changes?

- Do you have quantitative evidence—using region masks from PIE-Bench—that the learned intermediate state indeed corrupts editing-relevant regions more strongly than background?

---

### Note · Authors · 2025-11-13

I have read and agree with the venue's withdrawal policy on behalf of myself and my co-authors.